# Key Stakeholders’ Perspectives on Implementation and Scale up of HIV Self-Testing in Rwanda

**DOI:** 10.3390/diagnostics10040194

**Published:** 2020-04-01

**Authors:** Tafadzwa Dzinamarira, Collins Kamanzi, Tivani Phosa Mashamba-Thompson

**Affiliations:** 1Department of Public Health Medicine, School of Nursing and Public Health, University of KwaZulu-Natal, Durban 4001, South Africa; Mashamba-Thompson@ukzn.ac.za; 2College of Medicine and Health Sciences, University of Rwanda, Kigali P.O. Box 3286, Rwanda; ckamanzi@nursph.org; 3ICAP, Mailman School of Public Health, Columbia University, Kigali 2807, Rwanda; 4CIHR Canadian HIV Trials Network, Vancouver, BC V6Z 1Y6, Canada; 5Department of Public Health, University of Limpopo, Polokwane, Limpopo 0727, South Africa

**Keywords:** HIV self-testing, implementation, scale-up, key stakeholder

## Abstract

Introduction: The World Health Organisation recommends HIV self-testing as an alternative testing method to help reach underserved populations, such as men in sub-Saharan Africa. Successful implementation and scale-up of HIV self-testing (HIVST) in Rwanda relies heavily on relevant stakeholders’ involvement. We sought to explore HIVST key stakeholders’ perceptions of the implementation and scale-up of HIVST in Rwanda. Method: We conducted in-depth interviews with personnel involved in HIV response projects in Rwanda between September and November 2019. We purposively sampled and interviewed 13 national-level key stakeholders from the Ministry of Health, Rwanda Biomedical Center, non-governmental organizations and HIV clinics at tertiary health facilities in Kigali. We used a thematic approach to analysis with a coding framework guided by Consolidated Framework for Implementation Research (intervention characteristics, inner setting, outer setting, characteristics of individuals involved in the implementation and the implementation process). Results: Key stakeholders perceived HIVST as a potentially effective initiative, which can be used in order to ensure that there is an improvement in uptake of testing services, especially for underserved populations in Rwanda. The following challenges for implementation and scale-up of HIVST were revealed: lack of awareness of the kits, high cost of the self-test kits, and concerns on results interpretation. Key stakeholders identified the following as prerequisites to the successful implementation and scale-up of HIVST in Rwanda; creation of awareness, training those involved in the implementation process, regulation of the selling of the self-test kits, reduction of the costs of acquiring the self-test kits through the provision of subsidies, and ensuring consistent availability of the self-test kits. Conclusions: Key stakeholders expressed confidence in HIVST’s ability to improve the uptake of HIV testing services. However, they reported challenges, which need to be addressed to ensure successful implementation and scale-up of the HIVST. There is a need for further research incorporating lower level stakeholders to fully understand HIVST implementation and scale-up challenges and strategies to inform policy.

## 1. Introduction

Globally, it is approximated that only 79% of individuals who live with HIV are aware of their HIV status [1]. In Rwanda, the findings of a 2018–2019 national HIV survey in Rwanda indicated that 17% of adults living with HIV were unaware of their status [2]. By sex; 15% of HIV-positive women and 20% of HIV-positive men did not know their HIV status [2]. Improving the uptake of HIV testing services (HTS) remains one of the main strategies to combating HIV [3,4]. In 2016, the World Health Organization (WHO) provided the first global guidelines on HIV self-testing (HIVST), as an additional model for improving uptake of HTS [5]. Based on statistics from the WHO, 77 nations have adopted HIV self-testing policies, whereas several others are presently developing them [1]. WHO guidelines have been aimed at supporting the implementation and scale-up of ethical, effective, acceptable, as well as evidence-based approaches to HIVST [1]. Along with other sub-Saharan African countries, the Rwanda Ministry of Health in 2017 recommended the use of HIVST as an additional strategy [6]. 

HIVST refers to the process where an individual collects his own specimen, which could be blood or oral fluid, and thereafter carries out HIV testing and decodes the result, in most cases in a private setting [5]. HIVST has the potential to overcome some of the main barriers which are associated with the current testing models [7,8,9,10,11]. The notable barriers include stigma, discrimination, as well as non-confidential testing environments [7,8]. Acceptability studies have provided mixed findings. In South Africa, a very low rate of acceptability of HIVST (22%) was reported among conveniently-sampled adults [12]. Similarly, low acceptability (44%) was reported in China among men who have sex with men [13]. High acceptability rates have been reported among university students in the Democratic Republic of Congo (82%) [14]; men who have sex with men in Peru and Brazil (87%) [9] and a general adult population in western Kenya (94%) [11]. Globally, it has been documented in various studies that HIVST is acceptable in general populations [9,10,15,16,17]. A systematic review by Krause et al. presents evidence that HIVST is highly acceptable among key populations [8]. In Rwanda, a qualitative study on men’s perspectives towards HIVST revealed that men found HIVST acceptable; however, lack of awareness, cost of the kits, and concerns over potential social harm and possible adverse events were reported as potential barriers to uptake [18]. 

With a goal to end AIDS by 2030 [19,20]; this goal calls for strategic implementation and scale-up strategies that result in increased uptake HTS. Within SSA Africa, Kenya has effectively implemented guidelines on programmatic approaches to HIVST. The Kenyan guidelines for HIVST describe the package of support services, commodity management systems, the coordination mechanisms, quality assurance measures, and describing some of the monitoring and evaluation strategies [21]. These guidelines played a key role in ensuring effective implementation and scale-up of HIVST in Kenya. The relevance of stakeholders in the implementation of policies in every healthcare delivery cannot be over-emphasized. Evidence has demonstrated the important role of stakeholders in the successful implementation of health policies [22,23]. The implementation and scale-up efforts for HIVST in Rwanda will rely largely on informed strategies that ensure improved uptake [24]. Adequate involvement of all the relevant stakeholders is crucial for the overall success of the implementation and scale-up efforts. This study therefore sought to explore the perspectives of key stakeholders concerning the implementation and scale-up of HIVST in Rwanda. 

## 2. Materials and Methods

This study was conducted as part of a large study entitled: Adaptation of a Health Education Program for Improving Uptake of HIV Self-Testing among Men in Rwanda. The protocol for the main study is under consideration for publication elsewhere. 

### 2.1. Ethics

This study has been ethically reviewed and approved by four institutional review boards: the Rwanda National Ethics Committee (Approval number: 332/RNEC/201; May 29th 2019), University Teaching Hospital of Kigali Ethics Committee (Approval number: EC/CHUK/0111/2019; June 17th 2019), Rwanda Military Hospital Institutional Review Board (Approval number: RMH IRB/036/2019; July 12th 2019) and the University of KwaZulu Natal Biomedical Research Ethics Committee (Approval number: BE/280/19; June 24th 2019). Study participants were provided with an information sheet explaining the objectives of the study, and all participants signed informed consent forms prior to participation. 

### 2.2. Study Setting

Kigali Province is the capital city of Rwanda. It consists of three Districts, namely Gasabo, Kicukiro and Nyarugenge, 35 sectors, 161 cells, and 1183 villages in Kigali [25]. Kigali City houses all national-level stakeholders in the HIV program in Rwanda [26]. The overall HIV prevalence among adults in Rwanda is 3.0% [2]. Annual incidence of HIV among adults (defined as those aged 15–64 years) in Rwanda was 0.08% [2]. This corresponds to approximately 5,400 new cases of HIV annually among adults in Rwanda [2]. With a goal to end AIDS by 2030 [26], Rwanda is intensifying evidence-based interventions such as HIVST to further reduce HIV incidence. 

### 2.3. Study Sample

In the study, the sample consisted of 13 purposively-selected key stakeholders. In particular, the sample was drawn from the Ministry of Health, Rwanda Biomedical Center, non-governmental organizations and HIV clinics at tertiary health facilities in Kigali. 

### 2.4. Data Collection

We collected qualitative data from the key stakeholders using in-depth interviews. Interviews were conducted by trained researchers using an interview guide (Appendix A), which contained open-ended questions. Interviews were conducted between September 2019 and November 2019 in different settings where the stakeholders or public health officials serve. Interviews were conducted in English and in Kinyarwanda languages and continued until saturation was reached; when no additional information was emerging from the interviews [27]. Interviews conducted in Kinyarwanda were translated by a professional translator with back translation to ensure no loss of data. 

### 2.5. Data Entry and Analysis

The interviews conducted were audio-recorded and transcribed verbatim in Microsoft Word 2016. Verbatim transcription of all interviews, with study participant’s checking [27] to seek points of clarification in relation to issues arising from interviews, was performed to ensure the validity of the interviews. All interview transcripts were uploaded into NVivo v12 software (QSR International Pty Ltd., Melbourne, Australia) for analysis. Framework-based thematic analysis was performed by TD and CK, in parallel guided by the Consolidated Framework for Implementation Research (CFIR) [28]. The framework-based synthesis approach followed these steps: familiarization; identifying a thematic framework; indexing; charting; and mapping and interpretation. This approach has been applied in policy-related research questions [29]. This approach enabled domains identified in advance in the CFIR to be explicitly and systematically considered in the analysis, while also facilitating enough flexibility to detect and characterise issues that emerged from the transcripts [30]. First, the authors familiarized themselves with the content of the transcripts. Secondly, participants’ responses were coded into categories based on the CFIR domains, which were then grouped into nodes. Using the relationships module of NVivo, the nodes were grouped into similar concepts that reflect key stakeholders’ perspectives on implementation and scale-up of HIVST in Rwanda. Finally, mapping and interpretation of the themes and sub-themes was done. 

## 3. Results

In total, we interviewed 13 participants, including HIVST key stakeholders in Rwanda’s HIV programs. In this study, we defined HIVST stakeholders as professionals working within the Rwanda Ministry of Health and its partners responsible for the implementation of HIVST in Rwanda. The individuals formed part of the national-level technical working group for HIVST implementation with oversight of implementation and scale-up efforts nationally. The characteristics of study participants are outlined in Table 1.

Key stakeholders were all well-aware of HIVST intervention and perceived HIVST as a potentially-effective initiative which can be used in order to ensure that there is improvement in uptake of testing services, especially for underserved populations in Rwanda. 

### 3.1. Researcher, Stakeholder

“HIV self-testing basically refers to the process through which an individual who is interested in knowing their HIV status carry out the HIV test by themselves. They also interpret the result in private. It is just one of the ways through which individuals can get to know their HIV status after buying the self-test kits. It was introduced in order to address the challenges of stigma and confidentiality associated with routine provider initiated or voluntary counseling and testing. It offers the potential for HIV testing to reach more people than previously possible, including those who do not seek testing in our health facilities here”.

### 3.2. Supply Chain Specialist, Policy Advisor

“It is an additional or a new approach in Rwanda to boost the existing HIV testing services that we currently have. It is a screening test which detects HIV antibodies. Currently in our country, it is in the pilot phase, not yet well implemented in the whole country. We have kits distributed in different pharmacies in Kigali and those vendors explain to those who come to buy it on how they can use it accordingly”. 

## 4. Emerging Themes

Three main themes emerged: HIVST is a potentially effective initiative to improve uptake of HIV testing services; challenges hindering effective implementation; and potential strategies which can be adopted to ensure effective implementation and scale-up of HIVST. The emerging themes and sub-themes are presented in Table 2.

A detailed framework analysis of key stakeholders’ responses guided by the CFIR is presented on Appendix A. 

### 4.1. Theme 1

Key stakeholders perceived HIVST as an acceptable intervention with the potential to bridge the gap in the uptake of HTS. In addition to the general population, most stakeholders listed men, female sex workers, men who have sex with men, and the rick/famous people as potential groups that could benefit from HIVST initiative in Rwanda. 

#### 4.1.1. Program Manager, Policy Maker

“Thank you, so (HIV) self-testing is one of the approaches that we are using to make sure that people know their HIV status. It was started as an additional option for people who want to know their HIV status and especially those who are not willing to use conventional HIV testing methods which are facility-based. This was chosen as an approach that will be increasing the number of people who are aware of their HIV status”. 

#### 4.1.2. Researcher, Stakeholder

“I think it (HIVST) is a noble intervention, which will play a key role in ensuring that there is an improvement in the uptake of HIV testing services, more so among the people who are otherwise normally hard to reach. It comes with a number of advantages, which generally include privacy, convenience, confidentiality, and ease of use”. 

#### 4.1.3. Program Manager, Policy Maker

“HIV testing services are available, free of charge, and available to all public health facilities, but we have some groups who don’t reach those services for their own reasons, maybe self-stigma in the case of sex workers, maybe being too busy like men, or they are rich and famous. All these groups can now access [HIV] self-testing”. 

#### 4.1.4. Laboratory Specialist, Policy Maker

“Currently, we have only one kit available on the local market and at health facilities. OraQuick (manufactured) by Orasure Technologies. We performed in-house validations here at the reference lab. We obtained over 90% sensitivity for oral fluid. So as (HIV) program, we are confident this test can help improve our numbers on the first 90. More people can now get tested and interpret their own results. The only caveat is those results are not final; patients would still need to visit a health facility for retesting and confirmation”. 

### 4.2. Theme 2: Challenges Hindering Implementation and Scale up of HIVST

Key stakeholders alluded to challenges that were impeding the implementation and scale-up of HIVST in Rwanda. The participants presented a mix of intervention characteristics, outer and inner settings challenges. Most participants strongly felt the need to prioritize addressing these challenges before efforts to implement and scale-up HIVST are commenced in Rwanda. 

#### 4.2.1. Sub Theme 1: Lack of Awareness 

A general lack of awareness among the users of the kits emerged as a sub-theme. Stakeholders noted some concerns relating to lack of awareness of/on HIVST among the general population. Further, stakeholders noted that HIVST is still inaccessible to rural populations that are also resource-limited in terms of health facilities. Key stakeholders perceived that not much awareness has been created. They also noted that some of the people with access to the kits do not have adequate levels of education and knowledge on how to use them. 

#### 4.2.2. Researcher, Stakeholder

“Before the government decides to ensure that it is rolled up on a large scale, I think they need to ensure that people are aware of the kits. At the moment I think that not so many people know about them. Again, the government should work on the costs that are associated with the acquisition of the kits. At the moment, it is beyond the reach of most of the people who are the main targets.”

#### 4.2.3. Sub Theme 2: Cost of the Kits

Key informants perceived the costs of purchasing the kits as one of the main barriers to the effective implementation and scale-up of HIVST in Rwanda. 

#### 4.2.4. Health Care Provider, Policy Maker

“I think that the ministry of health is still facing different kinds of challenges, which needs to be addressed to ensure effective implementation of self-testing. For instance, the general lack of the guidance on HIV self-testing, and the lack of low-cost test kits and the systems to assess and regulate them, have been a key barrier to implementation. They need to be addressed effectively. The ministry of health should also adopt policy guidelines that inform the adoption of suitable HIVST test kits, taking into consideration who exactly we want to get tested.”

#### 4.2.5. Sub Theme 3: Results Interpretation

Key stakeholders noted their concerns on the interpretation of results following HIVST testing. Key stakeholders were of the perspective that reported cases of discordant results between the HIV self-test and repeat testing at a health facility were mainly user-based. Issues with the interpretation of results were revealed as key in these discrepancies.

#### 4.2.6. Program Manager, Policy Maker

“HIV self-testing is very new in Rwanda. So far, I only can think of two challenges. For instance, we have received cases of false positives and negatives. This creates a problem. Even though the cases are few, it is enough to raise concern as it may affect uptake. Once a bad message is passed about HIV self-testing giving incorrect results, it may lower uptake significantly. And we don’t want that. (…) Basically, it comes down to ensuring those that are selling in private pharmacies or distributing at our (government) facilities are trained so they can train the users and the issue of regulation of sale of these self-test kits.”

### 4.3. Theme 3: Strategies to Improve Implementation and Scale-up of HIVST in Rwanda

Key stakeholders presented their views on strategies which can be used to ensure successful implementation and scale-up of HIVST. These include the following: the creation of awareness; training those involved in the implementation process; regulation of the selling of the self-test kits; reduction of the costs of acquiring the self-test kits through the provision of subsidies; and ensuring consistent availability of the self-test kits.

#### 4.3.1. Sub Theme 1: Creation of Awareness 

The need for the creation of awareness on HIVST in order to ensure uptake was noted by most key stakeholders. Community mobilization strategies proposed by some stakeholders include decentralized campaigns, community-led advocacy through the monthly *umuganda* community meetings, and radio jingles. 

#### 4.3.2. Supply Chain Specialist, Policy Advisor

“The uptake of HIV self-testing is still very low, and as you may be aware, not much has been done to ensure that it is available to all the people in Rwanda, including those in the remote areas. So far, the focus has been only in Kigali City. There are also people who are still not aware of HIV self-testing. The government, therefore, needs to do much more to ensure that more people are made to be aware of HIV self-testing”.

#### 4.3.3. Program Manager, Policy Maker

“I recommend improving awareness, to encourage those groups who don’t get HIV testing and then to make availability of test kits at a low price.”

#### 4.3.4. Sub Theme 2: Training of the People Involved in the Implementation Process

Key stakeholders emphasized the need for providing training to all parties involved in the implementation of the HIVST intervention in order to ensure successful implementation. 

#### 4.3.5. Health Care Provider, Stakeholder

“The government should just ensure that it (implementation) is done in the right manner by training all the people involved, doing mass campaigns to ensure that more people are aware of the existence of the programs, and ensuring that there are proper distribution channels of the HIV (self) test kits”.

#### 4.3.6. Sub Theme 3: Proper Regulation of the Kits

A common theme, on the strategies to ensure effective implementation and scale-up of HIVST implementation, was the need to ensure proper regulation of the self-test kits. Stakeholders noted that currently in Rwanda HIV self-test kits are available for purchase online. Stakeholders perceived the need for ensuring guidelines are followed with regard to certification for use of these kits. 

#### 4.3.7. Laboratory Specialist, Policy Maker

“I also believe that proper regulation can play a key role in ensuring that some of the key challenges I have discussed earlier are addressed. We validated for use in Rwanda the test kits currently in circulation. We are also involved in ensuring quality assurance for new kit lots and surveillance for cases of false positive or negative results. We monitor those as well.”

#### 4.3.8. Sub Theme 4: Reducing the Costs of the Kits

Most of the stakeholders were of the viewpoint that the kits are currently going for 5000 RWF for one test, which is not within the reach of most of the people who are targeted. As a result, effective implementation and scale-up of the kits needs measures to be put in place to subsidize the costs of acquiring the kits. This will make the kits to be within the reach of the users.

#### 4.3.9. Researcher, Stakeholder

“I think the ministry should look at the processes that are currently being charged. The kit currently costs 5000 Rwandan Francs in pharmacies in Rwanda and I think this is too expensive. While the government is working to ensure that the kits are available in the entire country, they need to ensure that there are measures aimed at lowering the prices of the kits. For such an important intervention, it is reasonable to provide subsidies to cushion users”.

#### 4.3.10. Sub Theme 5: Ensuring Availability of the Kits

When it comes to effective implementation and scale-up of HIVST, key stakeholders perceived the need for the government of Rwanda to ensure that there are measures in place to ensure constant availability of kits. This means ensuring the availability of stock and proper distribution channels to improve uptake. 

#### 4.3.11. Program Manager, Policy Maker

“I think here the most important point is to see how we can increase kit availability by ensuring adequate stock levels at MPPD always, and appropriate space where the kits will be distributed. So far, we have them limited at pharmacies, a few health facilities and online purchases. To improve uptake when we scale-up, there is a need to have a wider range of distribution channels”.

## 5. Discussion

This study presents perceptions of HIVST key stakeholders on the implementation and scale-up of HIVST in Rwanda. Key stakeholders perceive HIVST to be a highly effective intervention for helping the underserved populations access HTS. This corroborates well with WHO recommendations on HIVST as an additional strategy to improve uptake of HTS [5]. Interventions aimed at improving uptake of HTS as an important step to attaining the UNAIDS 90-90-90 target of 2020 have been underscored in the Rwanda 2019–2024 Fourth Health Sector Strategic Plan [26]. Our findings reveal that key stakeholders perceive HIVST as an important gateway to realization of the UNAIDS 90-90-90 targets. Theme two identified key stakeholders’ perceived challenges for implementation and scale-up of HIVST; lack of awareness of the kits, high cost of the self-test kits, and concerns over results interpretation. Low awareness was mainly attributed to the intervention being relatively new in Rwanda and still in pilot phases. Theme three: key stakeholders’ perceived measures of what is necessary for the successful implementation and scale-up of HIVST in Rwanda included the creation of awareness, training those involved in the implementation process, regulation of the selling of the self-test kits, reduction of the costs of acquiring the self-test kits, and ensuring consistent availability of the self-test kits. Health education programs, community mobilization, development of HIVST country guidelines and provision of subsidies to cushion the cost of test kits would strengthen implementation and scale-up efforts for HIVST in Rwanda. 

In the current study, key stakeholders’ perceived HIVST as an auspicious intervention with the potential to bridge the current gap in uptake of HTS in Rwanda. This corroborates well with findings from a similar study conducted in South Africa, which demonstrated key stakeholders’ confidence in HIVST improving uptake of HTS in underserved population [31]. A recently published systematic review and meta-synthesis on men’s perspectives on HIV self-testing in sub-Saharan Africa recommended presented evidence of poor awareness but high acceptability of HIVST among men [32]. While this is the case; stakeholders expressed concerns that need to be addressed before effective implementation and scale-up can be achieved. Concerns cited by key stakeholders in the current study on the regulation of the sale of test-kits and results interpretation have been reported elsewhere [7,33,34,35]. Healthcare workers in Kenya concerned with challenges with test results interpretation recommended proper regulatory measures to be put in place prior to scale of HIVST intervention [7]. Similarly, healthcare providers in Kwa-Zulu Natal province in South Africa perceived issues with results interpretation as a potential challenge with HIVST implementation in South Africa [33]. A cross-sectional study on participants without prior experience with the HIV self-test revealed the most common interpretation error was incorrectly identifying a negative result as invalid [10]. Raising public awareness levels emerged as a key strategy to effective implementation and overall success of scale-up efforts in this study. Similar recommendations have been made elsewhere [31,32,36,37]. Key stakeholders in South Africa [31], researchers, academics, journalists, community advocates, policy makers and other key stakeholders in Nigeria [36] and reviews by Hlongwa et al. [32], Johnson et al. [37] all recommend need the improve public awareness on HIVST. Similar to the findings of the current study, ensuring that kits are affordable has been recommended by key stakeholders in South Africa [31,33] and potential users in Singapore [10]. 

The study has demonstrated the feasibility of HIVST implementation and scale-up in Rwanda from a key stakeholder’s perspective. There is a need to document HIVST guidelines and policies that define the supply chain, stakeholder roles and responsibilities, implementation strategy, quality assurance measures, and monitoring and evaluation strategies. Policymakers need to ensure that effective mobilization programs are designed to raise public awareness. Training for those involved in the implementation and subsequent step-down training will be key in the implementation and scale-up efforts of HIVST in Rwanda. 

A notable strength of the current study is that the majority of the key stakeholders were men. The current study is part of a larger study aimed at adaptation of a health education program for improving the uptake of HIVST among men. A limitation of this study was that only national-level key stakeholders residing in Kigali were enrolled, thus limiting the generalization of study findings to other settings. There is a need for further research incorporating lower-level stakeholders and to fully understand the challenges and inform policy. However, our sample was drawn from individuals who are responsible for the implementation of HIVST in the country, with knowledge on the status of the current implementation and scale-up efforts across the country. Finally, qualitative findings are highly subjective [38]. However, we used prolonged engagement [39,40] to ensure credibility and pilot testing of the interview guide [41,42] to ensure dependability. We enhanced the credibility and dependability of the study findings by following a rigorous inductive analysis and interpretation of the data. We also employed note-taking [42] and participant validation of transcripts [43] to enhance the credibility of the reported findings.

## 6. Conclusions

The current study findings demonstrate the confidence of key stakeholders in the Rwanda health system to effectively sustain the HIVST intervention. The concerns raised over factors with the potential to impede smooth implementation and scale-up should be addressed.

## Figures and Tables

**Table 1 diagnostics-10-00194-t001:** The presentation of key stakeholders by age, gender, highest qualification, number of years’ experience in the HIV response and role in HIVST implementation in Rwanda.

Participant ID	Age	Gender	Education Level	Years’ Experience in the HIV Response in Rwanda	Role in HIVST Implementation
#1	40	Female	Undergraduate	13	Program Manager, Policy maker
#2	30	Male	Undergraduate	5	Health care provider, policy maker
#3	41	Male	Undergraduate	10	Program Manager, Policy maker
#4	45	Male	Post-graduate	15	Program Manager, Policy maker
#5	42	Female	Post-graduate	15	Program Manager, Policy maker
#6	33	Male	Undergraduate	11	Supply Chain Specialist, Policy advisor
#7	29	Female	Undergraduate	5	Health care provider, policy maker
#8	44	Male	Doctorate	15	Researcher, Stakeholder
#9	50	Male	Doctorate	25	Researcher, Stakeholder
#10	48	Male	Undergraduate	15	Health care provider, policy maker
#11	49	Male	Post-graduate	15	Health care provider, policy maker
#12	37	Male	Doctorate	8	Laboratory Specialist, Policy maker
#13	39	Male	Doctorate	16	Researcher, Stakeholder

**Table 2 diagnostics-10-00194-t002:** Themes and sub-themes.

THEMES	SUB-THEME(S)
Theme 1: Potentially effective initiative to improve uptake of HIV testing services	Target people unwilling to use facility-based testing services
Theme 2: Challenges hindering implementation and scale up of HIVST	Poor awareness High cost of the kits Concerns on results interpretation
Theme 3: Strategies to improve implementation and scale up	Creation of awareness/knowledge of the intervention Training those involved in implementation Regulation of sell of self-test kits Reducing the costs of the kits through the provision of subsidies Ensure consistent availability of the self-test kits.

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
