# Peer review of "Key Stakeholders’ Perspectives on Implementation and Scale up of HIV Self-Testing in Rwanda"

_diagnostics, 2020, doi:10.3390/diagnostics10040194_

Round 1
Reviewer 1 Report
HIV is still one of the leading public threats in the world with approximately 37.9 million people living with infection despite the revolutionary advances in its prevention and treatment. Tafadzwa and colleagues report a stakeholders’ interview for the implementation and scale-up of HIV self-testing in Rwanda. The manuscript provided some inputs from 12 researchers and government officials and identified the main challenges of implementing a home-test in Rwanda. The topic is of great interest to the scientific community, healthcare providers, and governmental officials. The manuscript is worth attention, but the current form seems too immature for publication.
The major concern is that the authors didn’t include the most important stakeholder, patients or end-users. Feedback from patients is extremely important for policymakers to understand the unmet needs and the reason for the low acceptability of the current program. For example, if 5,000 RWF is too expensive (line 117, line 123), how much is affordable? If patients refuse to take the test (line 41), what measures can make it easier? What do patients think the easiest way to get the test kits? If many people are still unaware of the program (line 57), what are the reasons? Some stakeholders in the middle of the chain, e.g. pharmacies and retailers, may provide useful information from different perspectives. Basic information of the test needs to be described as HIV home testing itself is technically challenging. What tests are being implemented? Antibody-based or antigen-based? What is the clinical sensitivity? Who is the manufacturer? Were they performed at pharmacies or patients’ homes? What is the recommended protocol, how easy do they perform the test, and how are the results interpreted? Some interviewees mentioned false positivity and negativity (line 76, line 115). How significant are the false results? Were these false results due to the limitation of the test, the quality control of manufacturing, or the way the test is performed? Some typos.Line 42: were unaware know?
line 59: conveniently samples adults?
Line 94: Kigali City Province?
Line 112: was reache?
In sum, the manuscript focused on the interviews of the high-level management team may only provide limited information. Acquisition of feedback from other stakeholders especially from patients who will potentially use the test is essential for deeply understanding current challenges. Additionally, the authors need to describe and discuss the limitation from both the technical side and the policy side.
Author Response
Reviewer 1
Point 1: The major concern is that the authors didn’t include the most important stakeholder, patients or end-users. Feedback from patients is extremely important for policymakers to understand the unmet needs and the reason for the low acceptability of the current program. For example, if 5,000 RWF is too expensive (line 117, line 123), how much is affordable? If patients refuse to take the test (line 41), what measures can make it easier? What do patients think the easiest way to get the test kits? If many people are still unaware of the program (line 57), what are the reasons? Some stakeholders in the middle of the chain, e.g. pharmacies and retailers, may provide useful information from different perspectives.
Response 1: Thank you for this valuable comment. This study is a follow-up study to a study conducted to assess users’ perspectives on HIV self-testing. Results of this study have been published elsewhere. We have cited this study in the background; line 65-68; citation number 18. The current study focused on providers’ perspectives.
Point 2: Basic information of the test needs to be described as HIV home testing itself is technically challenging. What tests are being implemented? Antibody-based or antigen-based? What is the clinical sensitivity? Who is the manufacturer? Were they performed at pharmacies or patients’ homes? What is the recommended protocol, how easy do they perform the test, and how are the results interpreted?
Response 2: Thank you for this comment. We further review transcripts and cited a one interviewee on line 43-47 providing information on the test being implemented (name and manufacturer), sensitivity and current methods of distribution.
Point 3: Some interviewees mentioned false positivity and negativity (line 76, line 115). How significant are the false results? Were these false results due to the limitation of the test, the quality control of manufacturing, or the way the test is performed?
Response 3: One interviewee reported that the number of cases were few but still presented challenges to overall implementation of HIVST (line 82-85). Stakeholder perceived causes of incorrect results interpretation are presented in line 78-79. We have further provided a quote from the same interviewee (after the ellipsis, line 85-87) to report what the stakeholder perceived as potential causes of the false positivity and negativity.
Point 4: Some typos.
Line 42: were unaware know?
line 59: conveniently samples adults?
Line 94: Kigali City Province?
Line 112: was reache?
Response 4: Thank you for pointing out these typographical errors. We have corrected the typos highlighted and further reviewed the entire manuscript for any similar errors.
Point 5: In sum, the manuscript focused on the interviews of the high-level management team may only provide limited information. Acquisition of feedback from other stakeholders especially from patients who will potentially use the test is essential for deeply understanding current challenges. Additionally, the authors need to describe and discuss the limitation from both the technical side and the policy side.
Response 5: Thank you for this comment. We have further elaborated the limitation of this study being focused on high level stakeholders (line 194-197).
Reviewer 2 Report
Thank you for giving me the opportunity to review the article. The authors conducted a study on the key stakeholders’ perspectives on implementation and scale up of HIV self-testing in Rwanda. The topic is socially important, but additional information & discussion should be added before acceptance. I listed the comments below.
Comments:
Abstract:
The authors should add a sentence about future investigations with considering the limitations of this study.Methods:
The authors should clearly state about the definition of the “key stakeholder”. Did the authors refer the article #21 and #22 in the Reference section?Discussion:
Most of the key stakeholders were men. Therefore, the authors should discuss about gender difference. When discussing about it, they should state about the gender ratio in each role. I understood that it was difficult to generalize the results. However, the authors should discuss about the regional characteristics and difference from other regions in Rwanda.Author Response
Reviewer 2
Point 1: Abstract - The authors should add a sentence about future investigations with considering the limitations of this study.
Response 1: Thank you for this comment. This has been done. Line 38-38.
Point 2: Methods - The authors should clearly state about the definition of the “key stakeholder”.
Response 2: Thank you for this comment. A definition for key stakeholder is provided in line 140-143.
Point 3: Did the authors refer the article #21 and #22 in the Reference section?
Response 3: We presented earlier evidence that has demonstrated the important role of stakeholders in successful implementation of health policies in settings similar to Rwanda. We cited work by Aniteye et al. (Ghana) and Bennett et al. (Afghanistan, Bangladesh and Uganda) is well present in text (lines 76-78) and in the reference section.
Point 4: Discussion - Most of the key stakeholders were men. Therefore, the authors should discuss about gender difference. When discussing about it, they should state about the gender ratio in each role.
Response 4: Our main study is focused on adaptation of a health education program for improving uptake of HIVST among men in Kigali, Rwanda (line 86-88). We have added a line on the strength and limitation paragraph (line 192-194) to highlight how the gender composition of the study participants was a strength of the survey.
Point 5: I understood that it was difficult to generalize the results. However, the authors should discuss about the regional characteristics and difference from other regions in Rwanda.
Response 5: Thank you for this comment. We have provided more details on stakeholders' area (within Rwanda) of responsibility (line 142-143). Further we have indicated that the sample was drawn from individuals responsible for implementation of the intervention in the whole of Rwanda as a strength of the study (line 197-200).
Round 2
Reviewer 1 Report
The revised manuscript is much stronger than the first version. The reviewer's only minor concern is that the current title highlighting "key stakeholders" is too broad and misleading. If the goal of this manuscript is to report feedback from providers' perspectives, the title of the manuscript should be revised accordingly, like the citation #18 clearly shows "perspectives of Men attending Tertiary Institutions and Kimisagara Youth Centre in Kigali". Otherwise, the manuscript is good for publication.
Reviewer 2 Report
Thank you for giving me the opportunity to review the revised version of this article. The authors revised the manuscript appropriately according to the comments, and I thought that it can be accepted for publication in the journal.